# A Second Career for p53 as A Broad-Spectrum Antiviral?

**DOI:** 10.3390/v15122377

**Published:** 2023-12-03

**Authors:** Joe B. Harford

**Affiliations:** SynerGene Therapeutics, Inc., Potomac, MD 20854, USA; harfordj@synergeneus.com

**Keywords:** p53, TP53, gene therapy, tumor suppressor, innate immunity, host antiviral defenses, broad-spectrum antiviral

## Abstract

As the world exits the global pandemic caused by the previously unknown SARS-CoV-2, we also mark the 30th anniversary of p53 being named “molecule of the year” by *Science* based on its role as a tumor suppressor. Although p53 was originally discovered in association with a viral protein, studies on its role in preventing carcinogenesis have far overshadowed research related to p53′s role in viral infections. Nonetheless, there is an extensive body of scientific literature demonstrating that p53 is a critical component of host immune responses to viral infections. It is striking that diverse viruses have independently developed an impressive repertoire of varied mechanisms to counter the host defenses that are mediated by and through p53. The variety of ways developed by viruses to disrupt p53 in their hosts attests to the protein’s importance in combatting viral pathogens. The present perspective aims to make the case that p53 ought to be considered a virus suppressor in addition to a tumor suppressor. It is hoped that additional research aimed at more fully understanding the role of p53 in antiviral immunity will result in the world being better positioned for the next pandemic than it was when SARS-CoV-2 emerged to produce COVID-19.

## 1. The Role of p53 in Cancer and the Potential of *TP53* Gene Therapy

When p53 was named “molecule of the year” by *Science* in December 1993, the accompanying editorial stated “p53 and its fellow tumor suppressors are generating an excitement that suggests prevention now and hope for a cure of a terrible killer in the not-too-distant future” [1]. Although the “not-too-distant future” was not defined in the editorial, cancer is still with us, and researchers are still seeking to understand how p53 functions as a tumor suppressor and how best to translate this understanding to benefit cancer patients [2,3]. Contributing to the excitement around p53 back then were reports by two independent groups that simultaneously published papers in *Nature* [4] and *Science* [5] demonstrating that germline alterations in the *TP53* gene explained the propensity to develop cancers within what have come to be called Li-Fraumeni families. In the ensuing years, p53 solidified its reputation as a tumor suppressor with more than half of sporadic human tumors carrying some form of *TP53* alteration [6]. Even in tumors that have wild-type p53, dysregulations in pathways either upstream or downstream of p53 are frequently seen.

Based on p53’s tumor suppressor activity, *TP53* gene therapy to restore wild-type p53 has long been envisioned as a potential approach in cancer treatment [3]. Both viral and non-viral gene delivery technologies have been employed to achieve expression of exogenous wild-type p53 in tumors. One example of *TP53* gene therapy is found in the investigational agent termed SGT-53, which is an immuno-lipid nanoparticle comprising a cationic liposome targeted via a single chain antibody fragment recognizing the human transferrin receptor and carrying as its payload a plasmid expression vector encoding human wild-type p53 [7,8]. SGT-53 has been shown to be well tolerated by patients in two Phase I trials [9,10] and is now being assessed in patients with advanced pancreatic cancers in a Phase II trial (ClinicalTrials.gov Identifier: NCT02340117; accessed 13 November 2023).

## 2. The Function of p53 as a Pleiotropic Transcription Factor

One of the more useful and influential descriptions of cancer envisions the disease as having ten hallmarks that include alterations in metabolism, angiogenesis, genetic instability, immune evasion, cell death, replicative immortality, sustained proliferation, invasion/metastasis, inflammation, and the tumor microenvironment [11]. Each of these hallmarks have been individually linked to p53 [12], reinforcing the notion that p53′s ability to act as a tumor suppressor derives from its influence on multiple cellular pathways. We now know that p53 acts as a pleiotropic transcription factor in regulating a relatively large number of downstream target genes involved in cell cycle arrest, senescence, DNA repair, cellular metabolism, and cell death [13]. One function of p53 that has drawn considerable attention from researchers in the field of oncology over the last three decades is its role in responses to the DNA damage caused by ionizing radiation and certain chemotherapeutic agents that are used in the treatment of cancer patients. When cells are exposed to ionizing radiation, a frequently used stimulus of p53 activity, induction of about 500 different genes and repression of a smaller number of genes are observed [14,15]. Both ionizing radiation and certain chemotherapeutic agents can lead to double-strand breaks in cellular DNA (see Figure 1, upper panel), with p53 being a major determinant of this transcriptional response [16]. The fate of cells with damaged DNA can be either DNA repair and emergence from cell cycle arrest or irreversible growth arrest and death via apoptosis [2,3]. Although the cellular fate that prevails differs from one cell type to another, both of these DNA damage responses are regulated transcriptionally by p53. The familiar moniker “guardian of the genome” [17] derives from the DNA damage responses regulated transcriptionally by p53. Traditional cancer treatment modalities can induce DNA damage; thus, disruption of p53 tends to render tumors less sensitive to radiotherapy and certain chemotherapies [18]. The resultant radio- and chemoresistance of tumors contributes to the association seen between p53 disruption and poorer patient outcomes in a number of cancer types [19,20,21]. Restoration of p53 in tumors via gene therapy has long been seen as a way of sensitizing cancers to these traditional treatment modalities [22], and *TP53* gene therapy has been most often envisioned in the context of a combination therapy involving chemotherapy and/or radiotherapy. For example, the previously mentioned Phase II trial of SGT-53 (ClinicalTrials.gov Identifier: NCT02340117; accessed 13 November 2023) combines *TP53* gene therapy with the standard-of-care chemotherapeutic regimen for advanced pancreatic cancers.

As the age of immunotherapies in oncology has emerged, it has become clear that disruption of p53 can also affect the immune system’s ability to identify and destroy tumors [23,24], which is central to effective cancer immunotherapy. A current mainstay of immunotherapy against tumors is the use of immune checkpoint inhibitors (ICIs) [25,26]. The ability of tumor-targeted *TP53* gene therapy to augment the antitumor efficacy of ICIs has now been documented in several preclinical models [27,28,29,30,31]. As with the more traditional cancer treatment modalities, the response (or lack thereof) of tumors to immunotherapy is governed by an elaborate network of expressed genes, many of which are regulated directly or indirectly by p53, the pleiotropic transcription factor. Available data suggest that restoration of p53 via gene therapy alters the expression of a number of immunogenic markers on the surface of tumor cells and induces immunogenic cell death to render tumors more immunologically “hot”, i.e., responsive to immunotherapy (see Figure 1, middle panel) [27,28,29,30,31]. The pleiotropic nature of p53 as a transcription factor means that the loss or alteration of p53 would be expected to impact multiple cellular pathways involved in antitumor immune responses. It follows that any restoration of p53 function could move cancer cells back toward normalcy on multiple fronts, including not only their response to traditional therapies (i.e., radiotherapy and chemotherapy) but also to immunotherapy based on ICIs. Indeed, it appears that multiple p53-regulated genes participate in turning an immunologically “cold” tumor “hot” [27,28,29,30,31]. An attractive aspect of p53 restoration via gene therapy is that one can actually be agnostic as to which of the many p53-regulated genes are contributing to enhanced responses of tumors expressing exogenous p53. With p53 restoration comes all of the downstream genes and pathways that appear to play an important role in improving responsiveness to both more traditional therapeutic regimens and antitumor immunotherapies.

**Figure 1 viruses-15-02377-f001:**
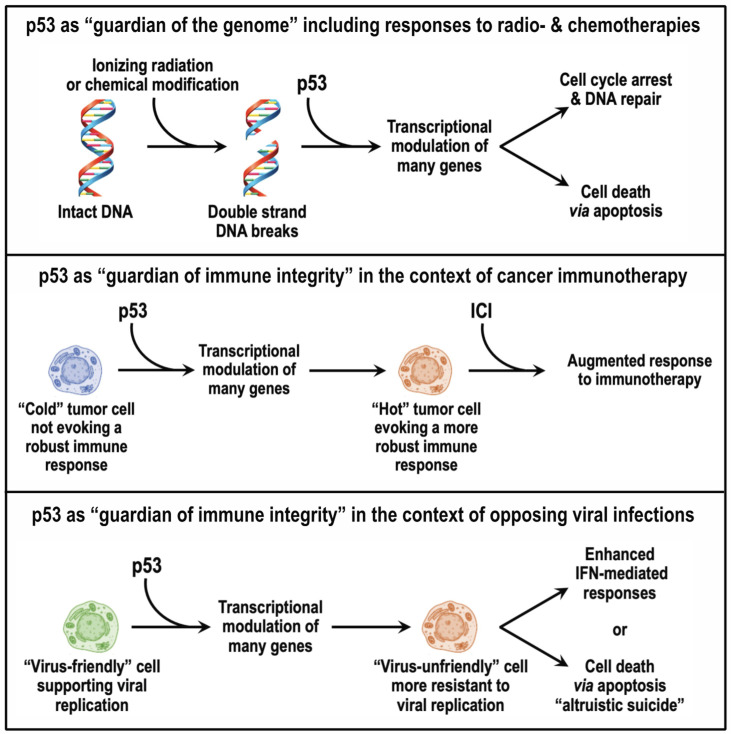
P53 serves as both “guardian of the genome” and “guardian of immune integrity”. Upper panel: Double-strand DNA breaks can occur when cells are exposed to environmental stresses, including ionizing radiation or DNA-modifying chemicals used in cancer treatments. In response to the DNA damage, p53 modulates the expression of many genes, including those involved in cell cycle arrest, DNA repair, and apoptosis. The fate of the damaged cell can either be an exit from cell cycle arrest with its DNA repaired or apoptotic death. In either case, the genome is effectively “guarded” [17]. Middle panel: Tumor cells with disruption of p53-mediated pathways can evade immune surveillance to qualify as an immunologically “cold” tumor, i.e., one that does not trigger a robust immune response and is refractory to cancer immunotherapy. Restoration of p53 (e.g., via gene therapy) can convert the “cold” tumor into an immunologically “hot” tumor that is more responsive to the immune checkpoint inhibitors (ICIs) used in cancer immunotherapy. Lower panel: Infecting viruses of diverse types seek to disrupt p53-mediated host pathways to create an environment that is more “virus-friendly”, i.e., favoring their replication and spread [32]. Maintaining or elevating p53 (e.g., via gene therapy) would be expected to enhance host antiviral responses mediated by interferons (IFNs) and/or trigger apoptosis (“altruistic suicide” [33]), thereby limiting viral propagation.

## 3. The Role of p53 in Host Defenses against Viral Infections

The same p53 that generated all the excitement three decades ago as a tumor suppressor is also a virus suppressor, and this fact should generate some excitement of its own. The discovery of p53 in 1979 was in the context of its association with a viral protein (SV40 large T-antigen) [2]. In addition, a linkage between p53 and infectious diseases was evident in early studies with p53 knockout mice. These mice developed tumors as anticipated, but about one-quarter of them died of unresolved infections prior to the appearance of tumors, suggesting that the loss of p53 resulted in a compromised immune system [34]. Subsequently, p53 knockout mice were shown to exhibit more severe disease after infections by influenza viruses [35]. In contrast, fibroblasts derived from “super p53 mice” having an extra copy of p53 exhibited higher resistance to the vesicular stomatitis virus [36], suggesting that even a rather modest increase in the *TP53* copy number can render cells more resistant to viral infections.

Viruses and their hosts do battle at the molecular level with the host using its gene products to detect viral infection and to suppress viral propagation. On the other hand, viruses employ diverse strategies to counter any mechanisms of their would-be host that impede viral replication. From the perspective of the virus, p53 is a defensive weapon of the host that needs to be neutralized or circumvented in some way. Why do viruses find it so necessary to oppose host p53? The answer appears to be in the many genes and cellular pathways that are downstream of p53. Triggering programmed cell death (apoptosis or ferroptosis) in response to “stresses” is considered a “canonical” function of p53 that is central to its tumor suppressor activity. Viral infections are also stressful, and apoptosis can be triggered by the incoming viral genomic DNA or RNA. This cell death in response to foreign viral nucleic acids may accomplish “altruistic suicide” that is a component of innate immunity [33], whereby the early death of the initially infected cells impedes viral replication and limits the spread of the virus to neighboring cells. In addition, the antiviral activity of p53 involves the regulation of diverse pathways producing interferons (IFNs). Regulation of type 1 IFNs is complicated and involves several p53-regulated cellular pathways, including increases in interferon regulatory factors IRF5, IRF7, and IRF9, each of which is transcriptionally regulated by p53 [37,38,39]. Signaling mediated by IFNs induces expression of a number of genes collectively referred to as IFN-stimulated genes (ISGs). These ISGs are varied, with some recognizing viral RNA replicative intermediates and others impeding viral replication by degrading viral RNA or inhibiting its translation to reduce the levels of viral proteins in the infected cell [40,41].

All available evidence suggests that p53 is a key participant in innate as well as adaptive antiviral immune responses [42,43]. Because p53 transcriptionally regulates many downstream genes=, viruses can negate a plethora of cellular defensive pathways by focusing their mechanisms for host evasion on p53. A virus with the ability to disable host p53 would reduce cellular levels of type 1 IFNs and thereby evade the range of its antiviral effects, including protection of uninfected bystander cells [40]. In analogy to its role as a tumor suppressor, it is likely there is more to p53’s viral suppressor activity than driving apoptosis and production of type 1 IFNs. Just as it has proven difficult to determine which of p53’s various activities is most crucial for its tumor suppressor functions [44], it will also likely prove challenging to distinguish which of the many p53-regulated genes and pathways are most critical for its viral suppressor activity. Nonetheless, the role of p53 in immune responses is generally now very well documented [23,43]. Disruption of p53 in tumors can affect the recruitment as well as the activity of myeloid and T cells, contributing to immune evasion by cancer cells. The growing appreciation of the role of p53 in immune responses has led to the suggestion that p53 be given the additional moniker “guardian of immune integrity” [32].

During evolutionary adaptation to their hosts, diverse viruses have developed a wide range of strategies to reduce p53 levels or attenuate its activity [43,45,46,47,48]. Many viruses encode subversive proteins that, in analogy to SV40 large T-antigen, directly interact with p53 to affect its activities, whereas other viruses use more indirect means of disrupting p53-mediated cellular processes. A detailed description of the various strategies by which viruses target and circumvent p53-mediated antiviral pathways is beyond the scope of this perspective. Suffice to say that, in order to thrive, viruses must subvert host defense mechanisms in which p53 participates, and some viruses employ more than one anti-p53 strategy. Dr. Arnold Levine, one of the discoverers of p53, has stated that “almost every successful virus has developed ways to inactivate p53” [43]. The very diversity of the repertoire of mechanisms used by viruses to disable p53 clearly attests to the importance of p53 in host antiviral defenses.

Among the viruses with countermeasures against p53 are the coronaviruses that include SARS-CoV-2, the etiological agent in the COVID-19 pandemic, which sadly has been a very successful virus in terms of its ability to infect humans. Two cell types that are part of the first line of defense against airborne viruses are lung alveolar macrophages and plasmacytoid dendritic cells, the latter being a major source of type 1 IFNs. Alveolar macrophages appear to be the first immune cells encountered by incoming respiratory viruses [49], and SARS-CoV-2 RNA has been detected in alveolar macrophages [50]. Several reports suggest that SARS-CoV-2 induces a relatively weak expression of IFNs in the lungs [50,51]. Use of type 1 IFNs against the coronaviruses SARS-CoV and MERS-CoV has been studied [52], and vapor inhalation of IFN-alpha in combination with ribavirin has been used against COVID-19 in China [53]. Although the interplay between coronaviruses and the type 1 IFN response is complicated [54], coronaviruses encode a protease with deubiquitinating activity that effectively reduces the level of p53 present in host cells following viral infection [46]. The weak IFN response in the lungs of COVID-19 patients would be consistent with the virus using this protease to reduce the p53 levels in the lung cells that would otherwise combat the infection via p53-regulated IFN pathways. Would alveolar macrophages be more effective in halting a SARS-CoV-2 infection if their p53 levels could be increased via *TP53* gene therapy? The investigational agent SGT-53 was designed to treat cancers that overexpress transferrin receptors. The rationale for repurposing this investigational agent to boost innate immunity against SARS-CoV-2 has been described [55] and is schematically depicted in Figure 2. Of particular relevance is the fact that both alveolar macrophages and plasmacytoid dendritic cells express transferrin receptors [56,57], and so both of these critical defensive cell types should be targeted by SGT-53 that uses cellular transferrin receptors to achieve *TP53* gene delivery.

## 4. The Next Pandemic

In the next pandemic caused by a hypothetical unknown virus that will emerge in the future, an approach based on enhancing innate antiviral immunity might prove advantageous over those strategies based on vaccines, since prophylactic vaccines require isolation and identification of the infectious agent and will take considerable time and effort to develop, test, and dispense. Moreover, success in producing antiviral vaccines is far from guaranteed—there is still no available HIV vaccine nearly 40 years after this virus was identified as the etiological agent in AIDS. The effectiveness of a vaccine is generally restricted to the virus for which it was designed (and perhaps some closely related variants), so vaccines tend not to be broad-spectrum protectors. In contrast, the innate immune system is poised to oppose any and all infectious agents whether known or unknown. Boosting innate immunity has the potential to protect against any virus that might come along in the future, even if that virus is being encountered by humans for the very first time.

In addition to the possibility of another viral pandemic beyond COVID-19 caused by an unknown virus, the U.S. Center for Disease Control and Prevention has identified certain viruses as potential weapons of mass destruction or agents with potential use in biological terrorism [58]. The very nature of bioterrorism would predict that the identification of a potentially deadly virus being employed by the terrorists would likely come only sometime after the first victims have been exposed. Warfighters facing an adversary using potentially deadly biological agents are also likely to be exposed to initially unidentified viruses. In either scenario, it would be desirable to have a life-saving option that could be more rapidly beneficial to virus-exposed individuals than is afforded by the lengthy path of identifying the culprit pathogen and developing a vaccine. The ideal therapeutic to address emerging or even newly created viruses that are released either unintentionally or intentionally by a malicious party engaging in biowarfare of bioterrorism would be a countermeasure that provides post-exposure protection against a broad spectrum of potential infectious agents, thereby allowing first responders and caregivers to remain agnostic regarding the precise nature of the infectious agent being employed. Increasing the level of cellular expression of a general virus suppressor like p53 may well be a means of achieving such broad-spectrum protection for individuals exposed to an unidentified viral threat agent.

Therapeutic targeting of p53 and certainly *TP53* gene therapy as an antiviral strategy are unorthodox notions. Antiviral drug development has historically been driven by the viral life cycle, with most antiviral agents being small chemical compounds (e.g., nucleoside analogs, protease inhibitors, integrase inhibitors, etc.) that directly target either viral replication per se or inhibit a cellular process required for viral replication [59,60]. Viruses replicate intracellularly and rely on the host’s synthetic machinery; thus, finding drugs capable of targeting viral replication without affecting critical processes of the host cell has proven challenging. The overlap between processes involved in viral replication and those used by our own cells can lead to untoward side effects of antiviral drugs. For example, nucleoside analogs are being used as antivirals, but these drugs can be toxic and/or mutagenic to host cells [61]. The product labels for approved antiviral drugs tend to list numerous potential serious side effects, and concerns about adverse events associated with these drugs are heightened when they are given to certain individuals (e.g., children, the elderly, pregnant women, immune-compromised individuals, or those with other pre-existing conditions). *TP53* gene therapy as an antiviral strategy differs from more traditional antiviral drugs in that it aims to boost innate immunity by increasing the level of a normal human protein that is already being expressed at some level in every cell of the body. Thus, restoring p53 expression in cells where its level or activity has been diminished by viral infection offers the possibility of broad-spectrum, pathogen-agnostic protection with minimal toxicity. It should be noted that very good safety profiles were observed in the Phase I SGT-53 oncology trials [9,10]. The extensive body of evidence of successful viruses with mechanisms to combat host p53-based defenses certainly makes exploring p53 maintenance and/or its restoration worthy of additional research in virology.

## 5. Summary and Perspectives

Not all cellular organisms face the threat of developing a tumor, but all such organisms do encounter infectious agents. It would therefore appear likely that the role of p53 (and closely related proteins p63 and p73) in innate immunity predates its better-known function as a tumor suppressor that earned it the distinction of “molecule of the year” back in 1993. Indeed, the DNA-binding domains that enable p53 and its relatives to function as transcription factors have been preserved over large evolutionary timeframes [2], suggesting functions in gene regulation that predate the need to suppress tumors. Nonetheless, in a word-association game, if participants are given the prompt of “p53”, the most common response would likely be “tumor suppressor”. A response of “virus suppressor” would be equally valid. The antiviral roles played by p53 in innate immunity have undoubtedly taken a back seat to its role in cancers in terms of research efforts. Since 1993, research on p53 has accelerated and the vast majority of this effort has focused on oncology. A crude bibliometric analysis using “p53” as a search term to probe for publications indexed in PubMed^®^ (https://pubmed.ncbi.nlm.nih.gov; accessed on 13 November 2023) revealed over 5000 p53 papers published in 2022 compared to under 1000 back in 1993. This analysis found a total of over 115,000 publications using “p53” as a search term. Of these papers, over 100,000 (~87%) were also hits for “p53 AND tumor”. This compares to under 10,000 papers (~8%) that were hits with “p53 AND virus”. It appears that research on the connection between p53 and viral infections may actually be diminishing over time. In 2022, over 5000 publications emerge from PubMed^®^ with “p53” as a search term and less than 300 (~5%) of these papers are hits with “p53 AND virus”. Whereas annual “p53” as well as “p53 AND tumor” publications have increased dramatically between 2000 and 2022, papers on “p53 AND virus” have decreased by ~40% over that same period. As we exit the global pandemic caused by SARS-CoV-2, a formerly unknown virus, it seems paradoxical that the interest in p53′s antiviral activity among researchers (as reflected in publication numbers) would be waning. Since the list of viruses that interfere with p53 in one way or another is long [45,46,47,48], restoration of p53 might very well be useful in aborting infections by a broad spectrum of viruses, including that newly emerging virus that will bring on the next pandemic. Our first line of defense against that pathogen (and the one after that one) will be innate immunity. There remains much to be learned by exploring the mechanisms by which p53 opposes viral infections and probing the variety of means used by different viruses to evade p53-mediated host defenses. This understanding may help equip the world to handle the next viral pandemic more effectively than it did the last. In the not-too-distant future, p53 may regain its title as “molecule of the year” on the basis of its second career as a broad-spectrum virus suppressor.

## Figures and Tables

**Figure 2 viruses-15-02377-f002:**
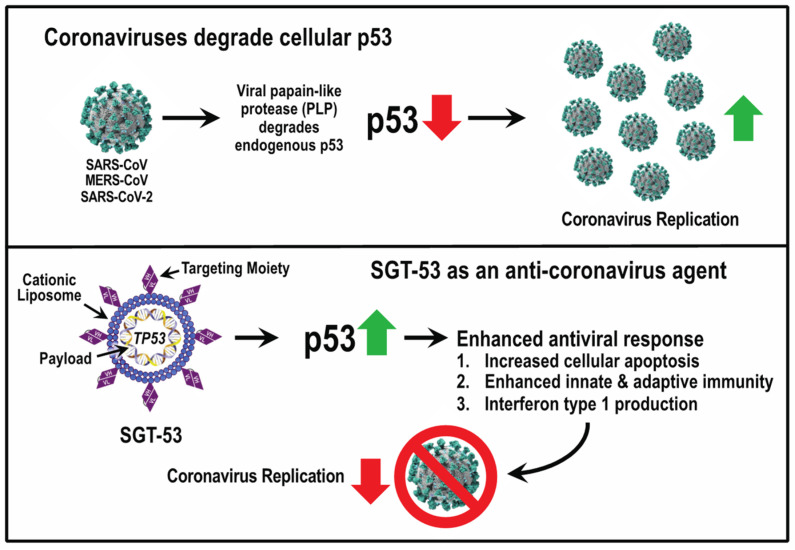
The potential use of TP53 gene therapy against coronaviruses infections. Coronaviruses, like other viruses, seek to thwart host defenses by decreasing p53 levels in the infected host cells. In the case of coronaviruses, this is accomplished via a protease that degrades p53 (upper panel). SGT-53 is a nanocomplex for TP53 gene therapy that was developed for oncology applications. SGT-53 has been proposed as an antiviral agent to combat coronaviruses like SARS-CoV-2 [55] by virtue of enhancing host innate immune responses regulated by p53 (lower panel). SGT-53 is a nanocomplex carrying a plasmid with the gene for wild-type human p53 within a cationic liposome that is decorated with a targeting moiety in the form of a single-chain monoclonal antibody fragment that recognizes the human transferrin receptor. SGT-53 was developed as a cancer therapeutic agent [7,8].

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
