# Peer review of "A Second Career for p53 as A Broad-Spectrum Antiviral?"

_viruses, 2023, doi:10.3390/v15122377_

Round 1

Reviewer 1 Report

Comments and Suggestions for Authors

In this perspective, the authors summarises p53's role in cancer and in the immune defense. Initially identified as an oncogene and then later corrected to being a tumor suppressor protein, it becomes more and more evident that p53 has additional activities. One of these additional activities is its role in the immune response. This role may become even more interesting to a wider readership, thinking about the recent pandemic, we all suffered from. In this light, this is the right time for such an perspective. Although a number of more comprehensive reviews on this action of p53 have been published in more recent years, a short and well written small summary of these activities as this one, written by Joe Harford, can raise the appetite and interest of a wider readership. Therefore, yes, this manuscript should be published in MDPI.

Nevertheless, I have a few remarks that should be addressed prior to publishing:

- Page 2, lane 63/64: The sentence "Cells respond...." is a repetition of the previous one and can be removed".

- Page 2, lane 65: The fate of(?) cells?

- Several statements are not supported by references e.g. page 2, lane 88 to 91 or page 2/3, lane 97 to 98.

- The expression "molecule of the year" is used a bit often in this short text

- The pubmed indexation of p53 in the Summary and Perspectives chapter is a bit too elaborate and should be condensed.

Reviewer 2 Report

Comments and Suggestions for Authors

In the perspective paper “A Second Career for p53 as A Broad-Spectrum Antiviral?”, the author summarized current knowledge about the role of p53 in cancer and host defenses against viral infections. The author also discussed that additional research is importance to understand the role of p53 as “virus suppressor” so that the next pandemic can be respond. The manuscript was very well written and the important publications including the latest ones were cited. Therefore, the manuscript is acceptable for this journal. However, I strongly recommend the following three points.

1.     Please remove the references in title of Figure 1.

2.     Please add explanation in the model diagram of Figure 1 about “Cold” tumor cell and “Virus-friendly” cell.

3.     Please add model diagram for lines 182 to 205.
